# Cabozantinib Following Immunotherapy in Patients with Advanced Hepatocellular Carcinoma

**DOI:** 10.3390/cancers14215173

**Published:** 2022-10-22

**Authors:** Michael H. Storandt, Jennifer J. Gile, Mathias E. Palmer, Tyler J. Zemla, Daniel H. Ahn, Tanios S. Bekaii-Saab, Zhaohui Jin, Nguyen H. Tran, Amit Mahipal

**Affiliations:** 1Department of Medicine, Mayo Clinic, Rochester, MN 55905, USA; 2Department of Oncology, Mayo Clinic, Rochester, MN 55905, USA; 3Department of Clinical Trials and Biostatistics, Mayo Clinic, Rochester, MN 55905, USA; 4Department of Hematology/Oncology, Mayo Clinic, Phoenix, AZ 85054, USA; 5Department of Oncology, University Hospitals Seidman Cancer Center and Case Western Reserve University, Cleveland, OH 44106, USA

**Keywords:** hepatocellular cancer, cabozantinib, immunotherapy

## Abstract

**Simple Summary:**

Management of hepatocellular carcinoma is a rapidly evolving field, with atezolizumab-bevacizumab recently becoming standard of care after showing survival benefit over sorafenib. However, all clinical trials evaluating drugs approved in the second line setting, such as cabozantinib, have been evaluated following progression on sorafenib, not immunotherapy. We sought to determine if cabozantinib is a viable option for patients after progression on immunotherapy. We conducted a retrospective analysis of patients seen at our institution who had disease progression on immunotherapy and subsequently received cabozantinib, reporting patient survival and tolerance of treatment. We found that patients had a median progression free survival of 2.1 months and median overall survival of 7.7 months, and most patients had a manageable side effect profile, suggesting that cabozantinib is a viable treatment option following progression on immunotherapy.

**Abstract:**

(1) Background: Cabozantinib, a multikinase inhibitor, is approved by the Food and Drug Administration (FDA) for the treatment of advanced hepatocellular carcinoma (HCC) following progression on sorafenib. Recently, atezolizumab plus bevacizumab has been approved in the first line setting for advanced HCC and has become the new standard of care. Whether cabozantinib improves outcomes following progression on immunotherapy remains unknown. We describe the clinical outcomes following treatment with immunotherapy in patients with advanced HCC who received cabozantinib. (2) Methods: We conducted a multicentric, retrospective analysis of patients with advanced HCC diagnosed between 2010–2021 at Mayo Clinic in Minnesota, Arizona, and Florida who received cabozantinib. Median overall survival and progression free survival analyses were performed using the Kaplan–Meier method. Adverse events were determined using Common Terminology Criteria for Adverse Events (CTCAE). (3). Results: We identified 26 patients with advanced HCC who received cabozantinib following progression on immunotherapy. Median progression free survival on cabozantinib therapy was 2.1 months (95% CI: 1.3–3.9) and median overall survival from time of cabozantinib initiation was 7.7 months (95% CI: 5.3–14.9). (4) Conclusion: The optimal sequencing of therapy for patients with advanced HCC following progression on immunotherapy remains unknown. Our study demonstrates that patients may benefit from treatment with cabozantinib following progression on immunotherapy.

## 1. Introduction

Sorafenib was the first systemic therapy receiving Food and Drug Administration (FDA) approval for advanced hepatocellular carcinoma (HCC) after it demonstrated improved median overall survival (mOS) when compared to best supportive care (10.7 months vs. 7.9 months, hazard ratio (HR) 0.69, *p* < 0.001), and remained the only first line therapy for greater than a decade [1]. Lenvatinib, an antiangiogenic tyrosine kinase inhibitor (TKI), was later evaluated in a non-inferiority study versus sorafenib in the first line setting and reached its primary end point of non-inferior mOS (13.6 months vs. 12.3 months, HR 0.92) [2]. Since, as our understanding of the pathogenesis of HCC has improved, treatment strategies and targets of intervention have begun to shift.

The pathogenesis of HCC is largely driven by chronic inflammation, often secondary to viral hepatitis or alcohol use, however, tumor cells are often able to evade immune recognition through generation of an immunosuppressive microenvironment, garnering interest in the role of immunotherapy in targeting HCC [3,4]. Recently, the IMbrave 150 study evaluated the combination of atezolizumab and bevacizumab versus sorafenib in the first line setting for treatment of advanced HCC, and demonstrated a mOS benefit (OS at 12 months 67.2% vs. 54.6%, HR 0.58), becoming the new standard of care for advanced HCC [5]. Since then, multiple studies have demonstrated survival benefit in the treatment of advanced HCC. The HIMALAYA trial, evaluating the combination of tremelimumab and durvalumab versus sorafenib demonstrated an OS benefit (HR 0.78), and durvalumab monotherapy was non-inferior to sorafenib (HR 0.86) [6]. A phase 3 study evaluating the combination of camrelizumab and apatinib, an anti-angiogenic TKI, demonstrated OS benefit when compared to sorafenib in the first line setting (22.1 months vs. 15.2 months, HR 0.62, *p* < 0.0001) [7]. RATIONALE-301 evaluated tislelizumab in the first line setting compared to sorafenib and demonstrated non-inferior OS (15.9 months vs. 14.1 months, HR 0.85, *p =* 0.0398) [8].

Currently, multiple systemic therapeutic options are FDA approved as second line agents for advanced HCC including cabozantinib [9], regorafenib [10], ramucirumab (for alpha-fetoprotein (AFP) >400 mg/mL) [11], and pembrolizumab [12,13,14]. However, all these therapeutic agents have been evaluated after progression on sorafenib. Currently, the best second line regimen is not established as there are limited trials examining treatment options following progression on immunotherapy [15,16].

Cabozantinib is a multikinase inhibitor, targeting vascular endothelial growth factor (VEGF), mesenchymal-epithelial transition factor (MET), and the anexelekto receptor tyrosine kinase (AXL) [17,18]. The CELESTIAL trial was a randomized, double-blinded, phase 3 trial which compared cabozantinib to placebo in patients with advanced HCC who had disease progression on sorafenib [9]. Cabozantinib demonstrated improved overall survival (10.2 months vs. 8.0 months, HR 0.76, *p =* 005) and subsequently received FDA approval for use in the second line setting.

As with other agents approved for second line therapy in advanced HCC, there is a paucity of data evaluating outcomes of patients who received cabozantinib following progression on immunotherapy in the first line setting. Considering increased use of immunotherapy in the first line setting following the results of the IMbrave 150 and HIMALAYA trials, it is imperative to evaluate optimal sequencing of treatment following progression on immunotherapy. In this study, we sought to characterize the outcomes of patients with advanced HCC who received cabozantinib after progression on immunotherapy.

## 2. Methods

We conducted a retrospective study of patients with radiologically and/or pathologically confirmed diagnosis of HCC treated at the Mayo Clinic Enterprise involving three sites at Rochester, MN, Scottsdale, AZ and Jacksonville, FL between 1 January 2010, and 31 December 2021. Patients and their clinical data were identified and obtained via an electronic medical record survey using key search terms. Demographic characteristics, including age at diagnosis, sex, body mass index (BMI), body surface area (BSA), clinical history, tumor stage and grade at diagnosis, and systemic treatments received, were recorded. The study was reviewed and approved by the Mayo Clinic institutional review board and deemed not to require informed consent.

The primary endpoints analyzed were progression-free survival (PFS) and OS following treatment with cabozantinib in patients who had previous disease progression on immunotherapy. PFS was defined as the time from initiation of cabozantinib until disease progression or death. OS was defined as the time from initiation of cabozantinib until death due to any cause. Secondary outcomes were objective response rate (ORR) and disease control rate (DCR). ORR was defined as achieving complete response (CR) or partial response (PR) per Response Evaluation Criteria in Solid Tumors (RECIST) version 1.1 [19]. DCR was defined as the proportion of subjects achieving CR, PR, or stable disease (SD) while on therapy. The distributions of OS and PFS were estimated using the Kaplan–Meier method. Median values were estimated along with 95% confidence intervals (CI). Two-sided log-rank testing was used to compare OS and PFS between subgroups. ORR and DCR were estimated within each subgroup and compared between groups using a Chi-Square test or Fisher’s Exact test for proportion. Adverse events were determined using Common Terminology Criteria for Adverse Events (CTCAE).

## 3. Results

One-hundred and thirty-one patients were identified who received cabozantinib for HCC, and among these, 26 received immunotherapy prior to cabozantinib. The median age of patients was 61 years (range, 39–81 years). Twenty-two (85%) patients were male and 20 (77%) were Caucasian. The median AFP at diagnosis was 29 U/mL (range, 1–71,929), with four patients having AFP ≥ 400 at diagnosis. Nineteen (73%) patients had cirrhosis at time of starting cabozantinib. Seventeen (65%) received prior embolization, 9 (35%) received prior ablation, and 8 (31%) received prior radiation therapy. Eighteen patients (72%) had Child Pugh score A at diagnosis. Baseline patient demographics and clinical characteristics are summarized in Table 1. With regard to prior immunotherapy treatment, 13 (50%) patients received atezolizumab/bevacizumab, 12 (46%) received nivolumab, and 1 (4%) received durvalumab prior to receiving cabozantinib.

Median PFS on treatment with cabozantinib was 2.1 months (95% CI: 1.3–3.9) (Figure 1). The DCR was 27% (7 patients) and the objective response rate was 4% (1 patient). Among 7 patients with disease control, 4 had Child Pugh score A at initiation of cabozantinib, 1 had B7, 1 had B8, and 1 was unknown. Six of these received cabozantinib in the third line and 1 received it in the second line. No patient had a complete response to cabozantinib. One patient with objective response had Child Pugh score A at initiation of cabozantinib and received it in the second line setting following first line atezolizumab plus bevacizumab. There were no statistically significant differences in PFS on cabozantinib therapy when stratified by hepatitis C infection status (*p* = 0.50), cirrhosis (*p* = 0.30), or when stratified by line of therapy that cabozantinib was received (2nd line vs. 3rd line and beyond, *p* = 0.39). When stratifying by Child Pugh status at time of initiation of cabozantinib, patients with Child Pugh class A liver function had a mPFS of 2.1 months (95% CI: 1.5–4.0) whereas those with Child Pugh class B liver function had a mPFS of 1.3 months (95% CI: 0.9–NE), although these were not significantly different (*p =* 0.55) (Figure 2). Median OS from initiation of cabozantinib therapy was 7.7 months (95% CI: 5.3–14.9) (Figure 3). At time of data collection, 19 patients (73%) had died from any cause, primarily due to disease progression.

Common adverse events reported while on cabozantinib included fatigue (50%), anorexia (35%), AST elevation (35%), diarrhea (31%), hypertension (27%), abdominal discomfort, dyspepsia, ALT elevation (23% each), stomatitis, weight loss, rash, peripheral edema (15% each), and constipation (12%) (Table 2). Seven patients experienced grade 3 or greater adverse events (one patient experienced two grade 3 or greater toxicities) including hypertension, diarrhea, anorexia, stomatitis, bowel obstruction, palmar-plantar erythrodysesthesia, and rectal abscess/fistula. Notably, 2 patients discontinued the medication due to side effects including blood pressure elevation and one patient who self-discontinued due to intolerance.

## 4. Discussion

In this study, we sought to determine PFS and OS of patients receiving cabozantinib who had disease progression on immunotherapy. Within our retrospective study, patients receiving cabozantinib after immunotherapy had mPFS of 2.1 months and mOS of 7.7 months, whereas in the phase 3 CELESTIAL trial following progression on sorafenib, patients receiving cabozantinib had a mPFS of 5.2 months and mOS of 10.2 months [9]. With regard to response, patients receiving cabozantinib following immunotherapy had a DCR of 27% versus 64% seen in the CELESTIAL trial following first line sorafenib [9]. A study conducted in Italy evaluated 96 patients receiving cabozantinib in the real-world setting, with 79.1% receiving cabozantinib in the third line setting, and 90% receiving sorafenib in the first line setting [20]. Among these patients, mPFS was 5.1 months and mOS was 12.1 months, which is comparable to the CELESTIAL trial. However, the study did not account for patients receiving prior immunotherapy.

While patients in the reported retrospective cohort receiving cabozantinib following progression on immunotherapy seemingly had poorer outcomes when compared to patients enrolled in the CELESTIAL trial receiving cabozantinib following progression on sorafenib, it is important to note that this retrospective cohort of patients was different than that enrolled in the CELESTIAL trial. While the CELESTIAL trial only included patients with Child-Pugh class A liver function who had received no more than 2 prior lines of systemic therapy, within our retrospective cohort, 7 patients (28%) had Child-Pugh class B liver function and 22 (84%) had received 3 or more prior lines of systemic therapy. These differences could significantly impact PFS and OS, and therefore, make direct comparison challenging. When looking at 7 patients with disease control, 4 had Child Pugh score A liver function and 6 received cabozantinib in the third line. Finkelmeier et al. conducted a retrospective analysis of patients who received cabozantinib for HCC, including 26% with Child-Pugh class B or worse liver function, and demonstrated mPFS of 3.4 months and mOS of 7.0 months, which is more consistent with that seen in our retrospective study, which supports the impact of liver function on patient outcomes [21]. Notably, the side effect profile for patients receiving cabozantinib was similar to previous studies suggesting that cabozantinib can be safely prescribed following treatment with immunotherapy.

Overall, this study does suggest a role for use of cabozantinib for HCC in the second line setting following progression on immunotherapy. As shown, adverse events were largely grade 1 or 2, and cabozantinib was generally well tolerated in patients who had previously received immunotherapy. Additionally, outcomes were comparable to those seen in other real world data that included patients with Child Pugh B liver function [21].

With atezolizumab plus bevacizumab now used as the first line standard of care, and durvalumab plus tremelimumab expected to received FDA approval in the near future, it will be important for future studies to evaluate other second line agents following immunotherapy. We have begun by evaluating cabozantinib, but future studies ought to evaluate other options, such as lenvatinib, regorafenib, or ramucirumab. Additionally, it is of utmost importance to compare second line agents in order to determine the ideal sequence of therapy going forward. Furthermore, important will be evaluating the combination of cabozantinib and other second line agents with immunotherapy. Cabozantinib was studied in combination with immunotherapy in COSMIC-312, which was an open label, randomized, phase 3 trial comparing the combination of cabozantinib with atezolizumab versus sorafenib in the first line setting, which ultimately demonstrated improved PFS (6.8 vs. 4.2 months, HR 0.63, *p =* 0.001) but failed to confer an OS benefit (15.4 months vs. 15.5 months, HR 0.90, *p =* 0.44) [22]. However, the combination of cabozantinib with immunotherapy has shown promising results in management of other malignancies, such as renal cell carcinoma, and therefore further evaluation is merited [23].

This study has several limitations including small sample size, diversity of immunotherapy agents used, and its retrospective nature. However, to the best of our knowledge, this is the largest retrospective study to report outcomes of patients with advanced HCC treated with cabozantinib following progression on immunotherapy. In this multicenter study, patients were included from different geographical locations accounting for diverse patient population. The results from this study provide data for treatment of patients with HCC after progression on immunotherapy.

## 5. Conclusions

With the approval of immunotherapy for treatment of advanced HCC in the first line setting, there has been increased ambiguity as to the ideal sequence of treatment in the second line setting and beyond. This retrospective study suggests a role for cabozantinib in patients who have had disease progression on immunotherapy, however, further studies comparing second line agents will be important going forward. In addition, administration of cabozantinib following immunotherapy was not associated with adverse toxicity profile.

## Figures and Tables

**Figure 1 cancers-14-05173-f001:**
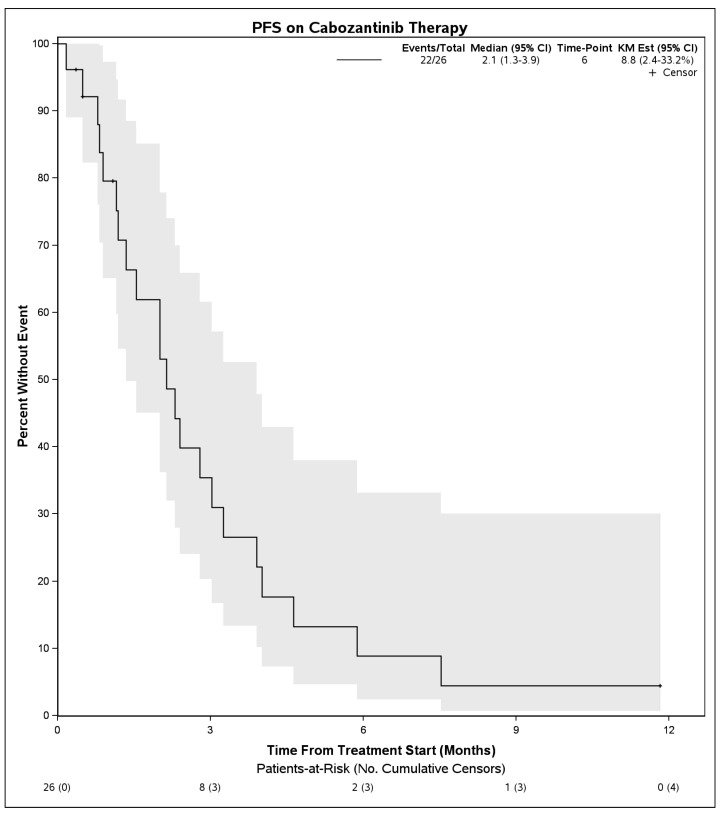
Progression free survival for patients with advanced hepatocellular carcinoma who received cabozantinib following progression on immunotherapy.

**Figure 2 cancers-14-05173-f002:**
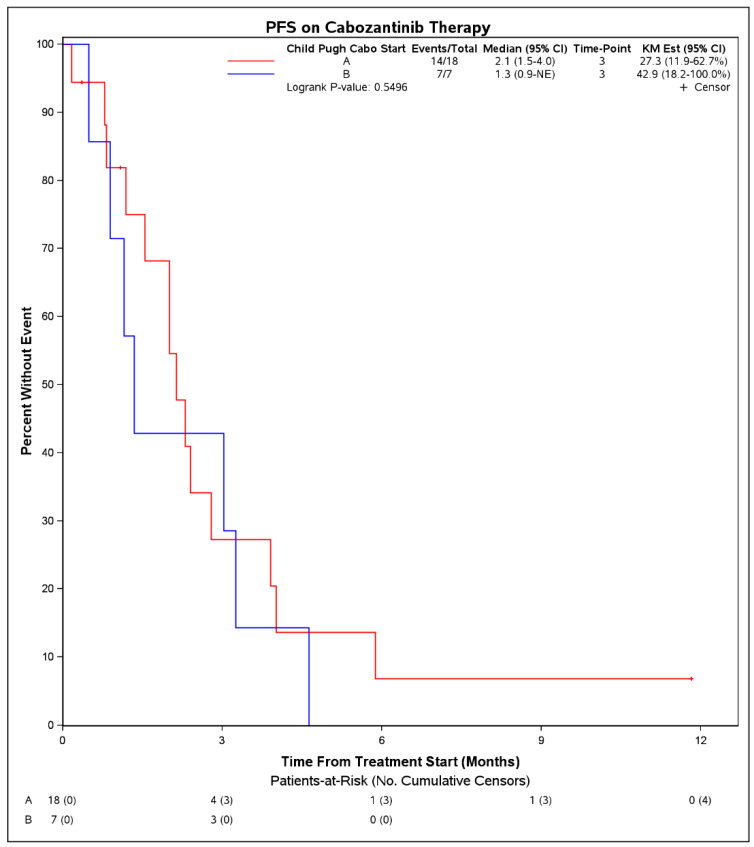
Progression free survival from initiation of cabozantinib therapy in patients with advanced hepatocellular carcinoma who progressed on immunotherapy, by patient Child Pugh score at time of initiation of cabozantinib.

**Figure 3 cancers-14-05173-f003:**
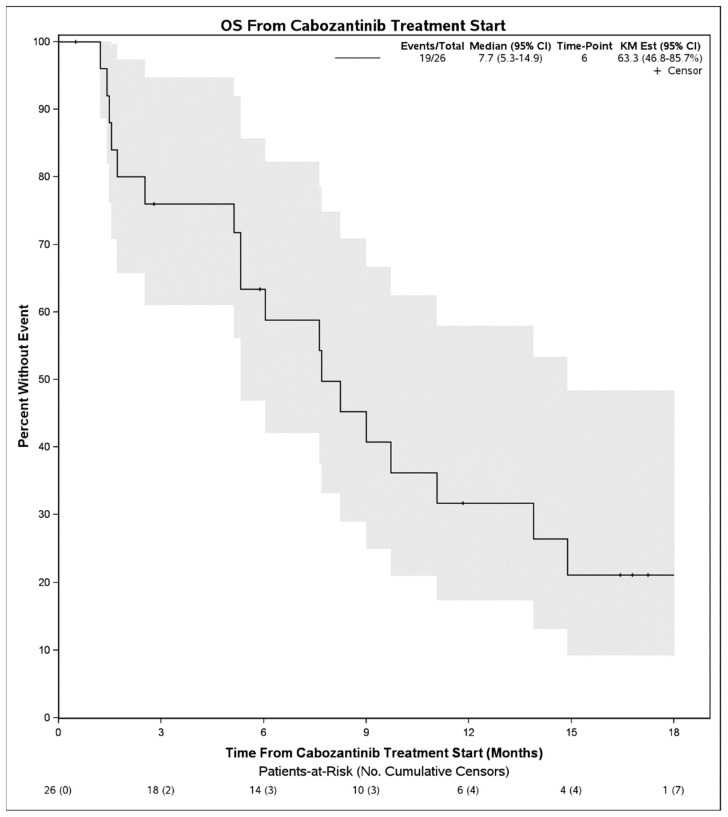
Overall survival from initiation of cabozantinib therapy in patients with advanced hepatocellular carcinoma who progressed initially on immunotherapy.

**Table 1 cancers-14-05173-t001:** Baseline patient demographics and clinical characteristics.

Characteristic	Patients N = 26
Median age, years (range)	61 (39–81)
Gender, N (%)	
Female	4 (15)
Male	22 (85)
Race, N (%)	
Caucasian	20 (77)
Asian	2 (8)
African American	4 (15)
Median BMI (range)	26.3 (19.7–42.3)
Median AFP at diagnosis (range)	29 IU/mL (1–71,929)
Received Prior Embolization, N (%)	17 (65)
Received Prior Ablation, N (%)	9 (35)
Received Prior Radiation Therapy, N (%)	8 (31)
Received Prior Systemic Therapy, N (%)	10 (39)
Cirrhosis Present, N (%)	19 (73)
Type II Diabetes Mellitus, N (%)	11 (42)
Hyperlipidemia, N (%)	10 (39)
Alcohol use, N (%)	8 (31)
Hepatitis C, N (%)	11 (42)
Non-Alcoholic Steatohepatitis, N (%)	2 (8)
Child Pugh Score at Diagnosis, N (%)	
Missing	1
A5	16 (64)
A6	7 (28)
B7	1 (4)
B8	1 (4)
Child Pugh Score at start of Cabozantinib Therapy, N (%)	
Missing	1
A5	11 (44)
A6	7 (28)
B7	3 (12)
B8	1 (4)
B9	3 (12)
Total Lines of Therapy Received, N (%)	
2	3 (12)
3	13 (50)
4	6 (23)
5	4 (15)
Line of Therapy Cabozantinib Received, N (%)	
2	4 (15)
3	18 (69)
4	4 (15)
Line of Therapy IO Received, N (%)	
1	10 (38)
2	15 (58)
3	1 (4)

**Table 2 cancers-14-05173-t002:** Adverse events experienced by patients while receiving cabozantinib.

Adverse Event	Any Grade N (%)	Grade 3 or 4 N (%)
Fatigue	13 (50)	0 (0)
Anorexia	9 (35)	1 (4)
AST elevation	9 (35)	0 (0)
Diarrhea	8 (31)	1 (4)
Hypertension	7 (27)	1 (4)
Abdominal discomfort	6 (23)	0 (0)
Dyspepsia	6 (23)	0 (0)
ALT elevation	6 (23)	0 (0)
Stomatitis	4 (15)	1 (4)
Weight loss	4 (15)	0 (0)
Rash	4 (15)	0 (0)
Peripheral edema	4 (15)	0 (0)
Constipation	3 (12)	0 (0)
Low Platelets	2 (8)	1 (4)
Dyspnea	2 (8)	0 (0)
Bowel obstruction	1 (4)	1 (4)
Palmar-Plantar erythrodysesthesia	1 (4)	1 (4)
Rectal abscess	1 (4)	1 (4)
Dry Skin	1 (4)	0 (0)
Hair loss	1 (4)	0 (0)
Dysgeusia	1 (4)	0 (0)
Headache	1 (4)	0 (0)
Back Pain	1 (4)	0 (0)
Insomnia	1 (4)	0 (0)
Flatulence	1 (4)	0 (0)
Confusion	1 (4)	0 (0)
Bilirubin elevation	1 (4)	0 (0)
Anemia	1 (4)	0 (0)

## Data Availability

The data presented in this study are available on request from the corresponding author. The data are not publicly available due to institutional restrictions.

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
