# Peer review of "Cabozantinib Following Immunotherapy in Patients with Advanced Hepatocellular Carcinoma"

_cancers, 2022, doi:10.3390/cancers14215173_

Round 1

Reviewer 1 Report

Although, multiple drugs have received FDA approval as second line drugs for treatment of advanced hepatocellular carcinoma (HCC), their efficacy has only been tested following progression on first line drug, sorafenib. Checkpoint blockade immunotherapy have now replaced this drug as the standard of care for HCC. Thus, it is imperative to test the efficiency of the FDA approved drugs as second line treatment options following progression on immunotherapy. This is necessary to design an optimal sequencing of therapy for patients with advanced carcinoma

With this study, the authors test the viability of multi-kinase inhibitor, cabozantinib, on disease free progression and overall survival of the patient, following immunotherapy.

The authors have done a thorough study with decent sample size of 26 HCC patients who received cabozantinib following immunotherapy as a second- or third-line drug. This study has a good mix of patients from different ethnicities with a good variation in the stage of carcinoma and other underlying health conditions. The article is written clearly and is easy to follow. Results are clearly represented figures and tables. However, certain points are not clear and I would like to suggest the following revisions:

1. Table 1 states that cabozantinib was used a 2nd, 3rd or the 4th line drug in patients following immunotherapy. It will be good to include whether any differences were observed between these patients and if there is any benefit in using cabozantinib as 2nd,3rd or 4th line of treatment.

2. Some patients received 4/5 lines of treatments, while cabozantinib was given as the 3rd line treatment for majority of the patients. How was the effect of these treatments evaluated on the disease free progression and overall survival of the patients.

Author Response

Comment: Table 1 states that cabozantinib was used a 2nd, 3rd or the 4th line drug in patients following immunotherapy. It will be good to include whether any differences were observed between these patients and if there is any benefit in using cabozantinib as 2nd,3rd or 4th line of treatment.

Response: In the results section we note that there were no statistically significant differences in PFS on cabozantinib therapy when stratified by line of therapy that cabozantinib was received (2nd line vs 3rd line and beyond, p = 0.39). This would suggest efficacy seen in earlier lines may still be seen when used later in therapy. Alternatively, due to limited number of subjects we did not observe statistically significant difference.

Comment: Some patients received 4/5 lines of treatments, while cabozantinib was given as the 3rd line treatment for majority of the patients. How was the effect of these treatments evaluated on the disease free progression and overall survival of the patients.

Response: When calculating PFS, this was defined as time from initiation of cabozantinib to time of progression on cabozantinib to maintain consistency. Similarly, for OS, time of initiation of Cabozantinib was used as a starting time. As noted above, we saw comparable progression free survival regardless of what line in which cabozantinib was received. There could certainly be differential effect based on lines of therapies but would require much larger sample size.

Reviewer 2 Report

General comment: In this manuscript, the authors evaluated safety and efficacy of carbozantinib treatment for tumor progression after immunotherapy. As the authors mentioned, there was paucity of data regrading this topic and thus seemed to be novel and important. The major weakness of this paper would be small sample size thus future study with large sample size would be required. The authors acknowledge this and successfully addressed why this study would be important though. This reviewers have minor comments below:

1. 1st paragraph of introduction is very well known to readers so it can be removed.

2. Page 3, method section, about RECIST: please reveal what did authors refer to. Old version? revised version? need reference for RECIST.

3. How many patients had AFP > 400 ? 

Author Response

Comment: 1st paragraph of introduction is very well known to readers so it can be removed.

Response: We have removed this paragraph as suggested.

Comment: Page 3, method section, about RECIST: please reveal what did authors refer to. Old version? revised version? need reference for RECIST.

Response: We have clarified as using RECIST version 1.1 and have added the appropriate citation.

Comment: How many patients had AFP > 400 ?

Response: This was present in 4 patients and has now been noted in the manuscript.

Reviewer 3 Report

 The authors of the manuscript touched upon important topic of the 2-nd line of HCC therapy after the using the gold standard Atezolizumab +Bevacizumab. It is necessary to develop clear criteria for choosing a drug as a 2-line therapy HCC.

Minor points:

1.       In the 1st sentence, you wrote, that, «liver cancer is the fifth most common malignancy globally», but by the link of references, I haven’t found this information. 

Middle points:

1.       If possible, could you please add some information about the etiology of HCC?

2.       I would appreciate, if you could explain, why you choose for your research Cabozantinib as 2-d line?  In NCCN Guidelines Regorafenib, Ramucirumab and Lenvatinib are also indicated as a 2-line therapy. Do you have statistic information of using this drugs as s 2-d line?

Author Response

Comment: In the 1st sentence, you wrote, that, «liver cancer is the fifth most common malignancy globally», but by the link of references, I haven’t found this information.
Response: We have now removed this paragraph altogether at the request of another reviewer.

Comment: If possible, could you please add some information about the etiology of HCC?
Response: We have added “The pathogenesis of HCC is largely driven by chronic inflammation, often secondary to viral hepatitis or alcohol use, however, tumor cells are often able to evade immune recognition through generation of an immunosuppressive microenvironment, garnering interest in the role of immunotherapy in targeting HCC,” to the introduction to briefly explain the pathogenesis of HCC and provide a primer to our discussion of immunotherapy in HCC.

Comment: I would appreciate, if you could explain, why you choose for your research Cabozantinib as 2-d line? In NCCN Guidelines Regorafenib, Ramucirumab and Lenvatinib are also indicated as a 2-line therapy. Do you have statistic information of using this drugs as s 2-d line?
Response: We have added the following statement to the discussion, “We have begun by evaluating cabozantinib, but future studies ought to evaluate other options, such as lenvatinib, regorafenib, or ramucirumab.” We note that while we
began with just cabozantinib, future studies ought to look at these other agents in the second line, as these findings will be equally important. Also, at our institution cabozantinib was much more frequently used compared to other therapies following progression on immunotherapies.
